# The Alteration of the Gut Microbiome during Ramadan Offers a Novel Perspective on Ramadan Fasting: A Pilot Study

**DOI:** 10.3390/microorganisms11082106

**Published:** 2023-08-18

**Authors:** YoungJae Jo, GyuDae Lee, Sajjad Ahmad, HyunWoo Son, Min-Ji Kim, Amani Sliti, Seungjun Lee, Kyeongnam Kim, Sung-Eun Lee, Jae-Ho Shin

**Affiliations:** 1Department of Applied Biosciences, Kyungpook National University, Daegu 41566, Republic of Korea; dudwo7573@knu.ac.kr (Y.J.); leegyuedae@gmail.com (G.L.); sajjadahmedbot1310@gmail.com (S.A.); thsgusdn2@knu.ac.kr (H.S.); tbd01188@knu.ac.kr (M.-J.K.); amanisliti25102020@gmail.com (A.S.); selpest@knu.ac.kr (S.-E.L.); 2Department of Food Science and Nutrition, College of Fisheries Science, Pukyong National University, Busan 48513, Republic of Korea; paul5280@pknu.ac.kr; 3Institute of Quality and Safety Evaluation of Agricultural Products, Kyungpook National University, Daegu 41566, Republic of Korea; kn1188@knu.ac.kr; 4Department of Integrative Biology, Kyungpook National University, Daegu 41566, Republic of Korea; 5NGS Center, Kyungpook National University, Daegu 41566, Republic of Korea

**Keywords:** Ramadan fasting, intermittent fasting, microbiome, short-chain fatty acids, *Lactobacillus*

## Abstract

An intermittent fasting regimen is widely perceived to lead to various beneficial health effects, including weight loss, the alleviation of insulin resistance, and the restructuring of a healthy gut microbiome. Because it shares certain commonalities with this dietary intervention, Ramadan fasting is sometimes misinterpreted as intermittent fasting, even though there are clear distinctions between these two regimens. The main purpose of this study is to verify whether Ramadan fasting drives the same beneficial effects as intermittent fasting by monitoring alterations in the gut microbiota. We conducted a study involving 20 Muslim individuals who were practicing Ramadan rituals and assessed the composition of their gut microbiomes during the 4-week period of Ramadan and the subsequent 8-week period post-Ramadan. Fecal microbiome analysis was conducted, and short-chain fatty acids (SCFAs) were assessed using liquid-chromatography–mass spectrometry. The observed decrease in the levels of SCFAs and beneficial bacteria during Ramadan, along with the increased microbial diversity post-Ramadan, suggests that the daily diet during Ramadan may not provide adequate nutrients to maintain robust gut microbiota. Additionally, the notable disparities in the functional genes detected through the metagenomic analysis and the strong correlation between *Lactobacillus* and SCFAs provide further support for our hypothesis.

## 1. Introduction

Ramadan is a period of fasting and spiritual reflection observed by Muslims, who comprise roughly 20% of the global population [1]. The holiday occurs during the ninth month of the Islamic calendar and is considered one of the Five Pillars of Islam, a set of fundamental religious obligations [1,2]. Exceptions to the fast are made for individuals who are sick, elderly, pregnant, nursing, or traveling. Children are also not required to fast until they reach puberty [1,3,4,5]. While those who are exempt from fasting are encouraged to make up the missed fasts at a later time, or to engage in acts of charity instead, Muslims generally abstain from food, drink, and other physical pleasures from sunrise to sunset during Ramadan [1]. This period of fasting occurs at regular intervals, leading to it occasionally being referred to as a form of intermittent fasting.

Intermittent fasting involves alternating periods of either consuming or abstaining from food [6]. The duration of the fast can be adjusted according to the individual’s objectives [7,8]. The primary purpose of intermittent fasting is weight loss, but further research is also being carried out to investigate its potential benefits for individuals with type 2 diabetes. For instance, a study published in the journal *Diabetes Care* suggests that intermittent fasting improves blood sugar control and diminishes the risk of heart disease in individuals with type 2 diabetes [9]. Another study by Suleiman et al. discovered that intermittent fasting enhanced insulin sensitivity and decreased blood pressure in type 2 diabetic patients [10]. However, despite their divergent objectives and approaches, a number of previous studies have overlooked the distinction between Ramadan and intermittent fasting and regarded them as equivalent [11,12,13,14,15,16]. Muslims undergo significant lifestyle changes during Ramadan, such as altered sleep patterns, water scarcity, and mealtimes. Under unfavorable conditions, such as during the hot summer, fasting can increase the risk of hypoglycemia, diabetic ketoacidosis, dehydration, and thrombosis [17,18,19,20]. Despite these potential risks, however, Ramadan has sometimes been believed to have benefits that are similar to intermittent fasting, as it does not appear to have any external effects on healthy individuals. However, such strict adherence to the regimen can disrupt the balance of microbes in the intestine, and dysbiosis has been linked to a wide range of diseases [21,22,23,24].

Dietary changes significantly modulate gut microbiota composition, an essential factor influencing host health. The gut microbiome, a complex community of trillions of microorganisms, responds dynamically to the nutrients we consume, underscoring its pivotal role in nutrition absorption, immune system modulation, and even behavioral influences. One such change that has been extensively studied is intermittent fasting, which has been shown to promote the growth of beneficial bacteria while reducing the abundance of pathogenic bacteria. This effect is exemplified by the proliferation of microorganisms, such as *Bifidobacterium* and *Lactobacillus*, which are known to produce SCFAs [13,25,26,27,28,29]. These compounds perform a crucial role in preserving gut homeostasis and have been associated with a decreased incidence of several diseases, including colon cancer, obesity, and type 2 diabetes [30]. In order to determine whether the Ramadan fast has equivalent health benefits to intermittent fasting, herein, we have conducted the first longitudinal investigation of the microbiome and SCFA levels both during and after Ramadan. Specifically, our findings provide unprecedented insights into the effects of Ramadan fasting in terms of gut health.

## 2. Materials and Methods

### 2.1. Subject Recruitment and Specimen and Data Collection

A total of 20 Muslims, comprising 15 Pakistanis and 5 Nigerians, were enrolled during the 2020 Ramadan period, from 23 April to 23 May. For 31 days of the Ramadan period, volunteers performed Ramadan fasting from sunrise to sunset (approximately 16 h per day). The study population comprised employees and students affiliated with Kyungpook National University, South Korea, from where the participants were selectively recruited. The study exclusively involved male participants who, based on their BMI, were categorized as having a healthy weight or being overweight. The comprehensive demographic data of the participants are detailed in Appendix A. Three exclusion criteria were employed in the recruitment of the participants: (1) individuals who had used antibiotics within the previous 3 to 6 months; (2) individuals who had been consistently taking probiotics for a minimum of 3 to 6 months; (3) individuals with medical conditions, including autoimmune diseases, irritable bowel syndrome, inflammatory bowel diseases, and diabetes. From all individuals, fecal samples were collected 5 times for 3 months. Sample collections were conducted three times during the Ramadan period (first, second, and last week), and once each at 4 weeks and 8 weeks after Ramadan, respectively. In this study, the samples acquired during the month of Ramadan were designated as R1, R2, and R4, while those obtained following Ramadan were labeled PR4 and PR8, respectively. To prevent contamination, we applied water-soluble recycling paper before collection, and a medical container (SPL Life Sciences, Gyeonggi-do, Republic of Korea) was used for storing the stool sample. After fecal sampling at their respective locations, all collected samples were immediately stored in a freezer at temperatures below −20 °C. They were then delivered to the lab within 24 h and stored at −80 °C until further experiments. In addition, the entire diet intake information was collected in the first, second, and last weeks of Ramadan, and in the fourth week after Ramadan. The acquisition of fecal samples was successful; however, several individuals had difficulty fulfilling the requirements of the PR8 survey pertaining to dietary intake due to personal reasons stemming from the COVID-19 pandemic. Caloric intake was calculated based on the dietary information on the MyFitnessPal website (https://www.myfitnesspal.com/ (12 July 2022)). Ethical approval for this study was obtained from the Bioethics Review Committee of Kyungpook National University (2020-0047). All volunteers provided written informed consent in accordance with the Helsinki Declaration prior to recruitment.

### 2.2. Measurement of Short-Chain Fatty Acid in Fecal Samples

To measure the levels of SCFAs in the fecal samples, we followed the methods by Han et al. [31]. First, to measure the dry weight of the fecal samples, 2 g was homogenized with 5 mL of acetonitrile and dehydrated using a high-speed vacuum centrifuge. The samples were then stored at −80 °C at a concentration of 2 mg/mL until further analysis. After being centrifuged at 4000× *g*, 20 °C for 10 min, the clear supernatants were collected for measuring the concentrations of SCFA. The supernatant was then combined with 3-nitrophenylhydrazine (3-NPH), N-(3-dimethylaminopropyl)-N′-ethyl carbodiimide (EDC), and an internal standard solution and heated to 40 °C for 30 min to derivatize the clear supernatant. The derivatized sample was pretreated for fecal SCFA analysis by adding formic acid to stop the reaction. The SCFAs were then determined using an Agilent 6460 Triple Quadruple Mass Spectrometer (Agilent Technologies, Santa Clara, CA, USA) and a 1260 Infinity UPLC system (Agilent Technologies, USA). The samples were separated using a Poroshell 120 SB-C18 column (Agilent Technologies, USA) with the mobile phase A being water containing 0.1% formic acid and mobile phase B being acetonitrile containing 0.1% formic acid, with a total flow rate of 200 L/min. The temperature of the thermostat column compartment was maintained at 60 °C throughout the analysis.

### 2.3. DNA Extraction and 16S rRNA Gene Amplicon Sequencing

Microbial DNAs from the fecal samples were extracted from approximately 200 mg of each stool sample using the QIAamp PowerFecal Pro DNA kit (Qiagen, Valencia, CA, USA) by following the manufacturer’s protocol. To construct sequencing libraries, the V4 to the V5 hypervariable region in the 16S rRNA gene was amplified using the 515 F (5′-barcode-GTGCCAGCMGCCGCGGTAA-3′) and 907 R (5′-barcode-CCGYCAATTCMTTTRAGTTT-3′) indexed reverse primers. The specific PCR conditions are described in a previous study [32]. Afterward, gel electrophoresis and a Qubit 2.0 Fluorometer (Thermo Fisher Scientific, Waltham, MA, USA) were used to confirm the quality and quantity of the library DNA. The library DNAs were then pooled equally and then purified using an AMPure XP bead (Beckman Coulter, Brea, CA, USA). The libraries were sequenced using an Ion Torrent PGM platform for 1250 flows with the Ion PGM^TM^ Hi Q Sequencing Kit (Thermo Fisher Scientific, USA).

### 2.4. Analysis of Bioinformatics 

Quantitative Insights into Microbial Ecology2 (QIIME2) pipeline (version 2020.11) was used for the data processing of the amplicon sequencing data [33]. According to individually designated barcodes, raw sequence reads were assigned to the samples. Samples with an average Phred quality score of less than Q30 (Phred Q30) were eliminated. A trimmed length of 250 bp was chosen to ensure both a quality score above Q30 and sufficient length for accurate taxa identification. and chimeric sequences were removed using DADA2 in order to yield amplicon sequence variations (ASVs) [34]. ASVs were selected using a naive Bayes QIIME2 classifier based on a distance of 0.01 (≥99% identity), and taxonomy assignment was performed using the SILVA database (silva-138-99-515-806-nb-classifier.qza). The sequences that were identified as either mitochondria, chloroplast, or were not assigned to any specific group, were excluded from the dataset. After this filtering process, all samples were standardized to the same sequencing depth of 2643 reads, leaving a total of 3587 ASVs.

### 2.5. Statistical Analysis

Among the five groups, the Friedman test was performed for multiple comparisons of the total calorie intake and nutritional components (protein, carbohydrate, and fat). In general, R studio software version 4.1.2 (https://www.rstudio.com/) was used to examine the gut microbiota of the participants. Individual alpha diversity indices (observed ASV and Shannon) were also computed using the Friedman test. Following multiple comparisons, Dunn’s test was applied as a post-hoc test, and statistical significance was accepted when the adjusted *p*-value was lower than 0.05. We employed Bray–Curtis dissimilarity for β-diversity to identify differences in composition among the groups. The distances were then visualized using two-dimensional principal coordinate analysis (PCoA), and statistical tests were conducted using permutational multivariate analysis of variance (PERMANOVA). In addition, the beta diversities of each group included their respective centroids, and box plots on two axes illustrate the dispersion. Significant differences in dispersion on the axes were simultaneously calculated using one-way analysis of variance (one-way ANOVA), and the *p*-values were adjusted by Holm correction. The statistical significance is represented by different letters. For the analysis of the microbial diversities, the “phyloseq” [35], “vegan” [36], and “ggplot2” [37] R packages were employed. The prevalence of ASVs in each group is shown as a heatmap using the “gplots” [38] R package according to the presence of ASVs in every sample. Taxonomic classifications were assessed at the phylum, family, and genus levels using the “phyloseq” [35] and “microbiome” [39] R packages. To be specific, in the pre-processing of the raw ASV data generated from QIIME2, taxa with less than 1% relative abundance and prevalence in the samples were excluded. The relative abundance at the phylum and family levels were generated as bar plots, and 47 genera are presented with their prevalence and phylum information on a heatmap. The Friedman test for multiple comparisons and the Wilcoxon test with Benjamini–Hochberg (BH) adjustment were used to discover the taxa with statistically significant differences among the groups. In addition, accessory and unique bacteria exclusively detected in the post-Ramadan groups were identified by core microbiome analysis using the “eulerr” [40], “microbiome” [35], and “microbiomeutilities” [41] R packages. To better understand, unassigned ASVs were removed and ASVs precisely assigned to genera were adopted. For the prediction of the microbial functional gene in each group based on 16S rRNA gene sequencing data, the ASVs were aligned to reference sequences from Phylogenetic Investigation of Communities by Reconstruction of Unobserved States (PICRUSt2) version 2.4.1 [42]. We then conducted a correlation analysis to determine the association between SCFAs and the gut microbiome.

## 3. Results

### 3.1. Participant and Nutritional Characteristics during Ramadan

A total of 20 participants were requested to not control their diet, and entire regimens were conducted ad libitum. Their personal information, including age, gender, height, weight, BMI (body mass index), and mealtime, is summarized in Appendix A. The dietary questionnaire and caloric information are attached in Appendix A. The participants ingested significantly more total calories following Ramadan (Dunn’s, * *p* < 0.05, ** *p* < 0.01; Figure 1A). Specifically, except for R2, the volunteers did not show a significantly different proportion of carbohydrate intake between the Ramadan and post-Ramadan period (Dunn’s, *** *p* < 0.001, **** *p* < 0.0001; Figure 1B). In addition, the calorie consumption of fat and protein exhibited an identical pattern to that of carbohydrate intake. However, protein consumption in R2 was significantly higher than in the other groups, while the pattern of fat consumption was the opposite (Dunn’s, ** *p* < 0.01, *** *p* < 0.001, **** *p* < 0.0001; Figure 1C,D). The specific caloric intake information is attached in Appendix A.

### 3.2. Ramadan Fasting Drives Substantial Change in Fecal Microbial Diversity

To discriminate how the fecal bacterial structures of the participants were different during Ramadan fasting, we monitored the patterns of the alpha and beta diversities based on the amplicon sequencing of the 16S rRNA gene. PCoA of Bray–Curtis dissimilarity revealed distinct microbial communities between the R (R1, R2, R4) and PR (PR4, PR8) groups. A total of five groups, including R1, R2, R4, PR4, and PR8, had significant differences in the microbial distance (Adonis, *p* = 0.05, Figure 2A). In addition, statistical analysis showed that the dissimilarity in the gut microbiome of Muslims during Ramadan was significant (Holm correction, *p* < 0.05, Figure 2A). The results demonstrate that Ramadan fasting affects the alpha diversity, the diversity of taxonomically different unicellular microorganisms, as evidenced by the distinct pattern observed. The observed ASV index was confirmed to be 70.5 as the median value of the microbial diversity in R1 and showed a tendency to decrease slightly during Ramadan. However, the post-Ramadan groups PR4 and PR8 revealed the opposite pattern. In the eight weeks following the conclusion of Ramadan, the observed ASVs of the volunteers increased considerably to 113 and 155.5, respectively. The changing pattern of the Shannon index was parallel to the observed ASV. The Shannon value of the individuals declined progressively as the four weeks of Ramadan continued but increased significantly to 4.49 once Ramadan was over (Dunn’s, * *p* < 0.05, ** *p* < 0.01, *** *p* < 0.001, **** *p* < 0.0001; Figure 2B). Furthermore, we detected a consistent loss of intestinal microbial ASVs in the Ramadan groups (Figure 2C).

### 3.3. Ramadan Fasting Causes Decline of Specific Taxa and Metabolic Functions

Contrary to the change in microbial diversity, no statistical significance was found in the relative abundance at the phylum and family levels. The microbial compositions of the individual Muslim fasting participants showed varying proportions of *Bacteroidota*, *Firmicutes* at the phylum level, and *Lachnospiraceae*, *Prevotellaceae*, *Ruminococcaceae*, and *Bacteroidaceae* at the family level. Interestingly, *Spirochaetota*, including several pathogenic species, were detected in a small proportion of the microbial community’s participants (Appendix A). Nonetheless, after Ramadan, the population of this phylum dropped substantially. In addition, throughout the observation period, during and after Ramadan fasting, the majority of the participants’ bacterial compositions consistently comprised *Prevotella*, *Blautia*, *Faecalibacterium*, and others (Figure 3A). Among the entire gut microbial community genera, the *Coprococcus* and *Lachnospiraceae NK4A136 group* showed significant reductions during Ramadan (BH, * *p* < 0.05; Figure 3B). Via core microbiome analysis, several ASVs belonging to beneficial bacteria were only identified in the post-Ramadan population. *Bacteroides*, *Lactobacillus*, and *Sutterlla* were observed in PR4, *Ruminococcaceae UCG-005*, and *Agathobacter* were found in PR8, and *Fusicatenibacter* and *Lachnoclostridium* were observed in both PR4 and PR8 (Figure 3C). Moreover, we implemented PICRUSt2 to investigate the significant change in the metagenome functional contents, and the top 10 of increased and decreased Kyoto Encyclopedia of Genes and Genomes (KEGG) orthologs ranked as level 3 were obtained. According to the results of the DESeq2 statistical analysis, the cystathionine beta-synthase (*CBS*, K01697) gene in R4 was the only functional characteristic revealed to be significantly enriched when compared to the PR8 group (FDR adjusted *p* = 0.02; Figure 3D).

### 3.4. Significant Alteration in Taxa from Ramadan Are Associated with Reduction of Short-Chain Fatty Acids and Microbial Diversity

The computed SCFAs, including acetate, butyrate, and propionate, showed a consistently decreasing tendency during Ramadan fasting, which lasted until PR4. However, these metabolites showed a recovery pattern in PR8 (Figure 4A). To examine the association between the gut microbiota and SCFAs, particular microorganisms (*Coprococcus* and *Lachnospiraceae NK4A 136 group*) and a functional gene (Cystathionine β synthase) that changed significantly before and after Ramadan, and observed taxa in only the post-Ramadan groups (*Bacteroides*, *Lactobacillus*, *Sutterlla*, *Fusicatenibacter*, *Lachnoclostridium*, *Ruminococcaceae UCG-005*, and *Agathobacter*), SCFAs (acetate, butyrate, and propionate) and alpha diversity indexes (observed ASV and Shannon) underwent correlation analysis. Remarkably, a strong association between *Lactobacillus* and SCFAs was discovered, and for a better understanding, the association is described using scatter plots with linear correlation (Rho = 0.64, *p* = 0.0024 for acetate; Rho = 0.62, *p* = 0.0034 for butyrate; Rho = 0.71, *p* = 0.0004 for propionate, respectively; Figure 4B,C).

## 4. Discussion

In recent decades, Ramadan fasting has been equated with intermittent fasting, and thus, has been promoted for its potential health benefits [43,44,45]. Preliminary studies investigating the potential relationship between Ramadan fasting and the gut microbiome have employed either a cross-sectional design comparing a fasting group (those participating in Ramadan fasting) with a group refeeding after Ramadan [11,12,13], or have used a longitudinal design monitoring changes in the gut microbiota over the course of the Ramadan month [14,15,16]. To the best of our knowledge, this is the first study to track changes in the gut microbiome throughout and then after Ramadan. Moreover, our academic achievements provide insight into the potential effects of Ramadan fasting on the gut microbiome and associated metabolic changes, highlighting the need for a more comprehensive understanding of this practice. In this study, we followed the gut microbial structure of individual participants for 12 weeks, including the mid- and post-Ramadan periods. The study cohort consisted of a single gender and a narrow age range, which helps to minimize confounding variables for gut microbiome exploration and increases the reliability of the results. We also recorded the food and nutrient intake of the study volunteers during the period of Ramadan fasting. Except for the total caloric, carbohydrate, and protein intake at a specific point, the ratio of ingested nutrient contents did not show a significant change, and the total amount of calories consumed after Ramadan was slightly higher. The caloric intake during Ramadan, which constitutes 75% of the caloric intake after Ramadan, conflicts with the primary goal of weight loss in intermittent fasting. In addition, equating Ramadan fasting with intermittent fasting requires consideration of the study participant characteristics. A review study conducted by Stephanie et al. [46] revealed that most of the individuals participating in the intermittent fasting study were obese, with an initial BMI above 30 kg/m^2^, with an average BMI of 34.7 kg/m^2^ across 16 studies. Conversely, the participants in our study were physically and mentally healthy individuals who could adhere to periodic fasting for a month, with an average BMI of 24.78 kg/m^2^, and only one participant had a BMI above 30 kg/m^2^ (Appendix A). As a result, it is challenging to identify the advantages of intermittent fasting in Ramadan fasting, which was conducted for subjects without the need for or goal of weight loss.

While fasting regimens varied among the participants, there was no significant difference in the beta diversity over the same period. After Ramadan, however, we detected a considerable microbial dissimilarity and an increase in the alpha diversity. In addition to the alpha diversity, the difference in ASV prevalence by diet may be interpreted as a result of ASV loss due to irregular eating habits and inadequate nutrition delivery during Ramadan. These findings suggest that the perturbation of microbial diversity during Ramadan warrants further investigation to determine the potential influence of fasting on the gut microbiome.

The comparison of shifts In the relative abundance of genera and an analysis of the core microbiome allowed for a better understanding of the diversity, and the loss of ASVs confirms our assumptions. Although the microbial composition of the total groups mostly consisted of *Prevotella*, *Blautia*, and *Faecalibacterium* regardless of regimen, we found a notable drop in the relative abundance of the *Coprococcus* and *Lachnospiraceae NK4A136* groups following Ramadan fasting. The genus *Coprococcus*, comprising anaerobic cocci, has been extensively studied for its potential to promote microbial homeostasis in the host through synergistic interactions with beneficial endogenous microbiota [47]. Additionally, *Coprococcus* has been shown to exert antipathogenic effects through mechanisms such as competitive exclusion, the enhancement of epithelial barrier function, and the production of antimicrobial compounds [48]. Similarly, the *Lachnospiraceae NK4A136 group*, a member of the *Lachnospiraceae* family, plays a significant role in the intestinal environment. For the most part, of those beneficial effects, the strong potential of the *Lachnospiraceae NK4A136 group*, which is producing SCFAs, including butyrate and acetate, has been made apparent by previous studies [49]. The loss of beneficial bacteria is reflected in trends seen in key microbiome assays. A number of taxa, including *Bacteroides* [50], *Lactobacillus* [51], *Sutterella* [52], *Fusicatenibacter* [53], *Lachnoclostridium* [54], *Ruminococcaceae UCG-005* [55], and *Agathobacter* [56], which were present only in the post-Ramadan group, have been studied for their role in SCFA production. Moreover, *Lactobacillus* has been found to influence immune system regulation and metabolic function, along with other *Clostridia* families, such as *Fusicatenibacter*, *Lachnoclostridium*, *Ruminococcaceae UCG-005*, and *Agathobacter* [57]. Notably, the role of *Sutterella* in human health has been debated, but its recent identification as a potential regulator of blood sugar levels aligns with the findings of this study [58].

The gut microbiome underwent significant changes in the predicted functional genes following Ramadan fasting, as demonstrated by the log2 fold change in the metabolic profiles via PICRUSt2 analysis. This analysis revealed extremely low levels of *CBS* in the microbial community, an enzyme coded by the *CBS* gene that is involved in the production of hydrogen sulfide (H_2_S). Previous research in animal models has shown that high concentrations of H_2_S in the gut are associated with increased microbial abundance, while low levels of H_2_S have been linked to inflammatory bowel disease and other gastrointestinal disorders [59]. H_2_S is thought to have various physiological roles in the gut, including in the regulation of blood flow, protection against oxidative stress, and the modulation of the immune system [60,61]. Therefore, it is possible that the decreased levels of H_2_S in the intestine resulting from Ramadan fasting may have various negative impacts on a person’s health.

With these alterations in the microbiome taxa and function, we examined the relationship between the microbiome and SCFAs. SCFAs, including butyrate, acetate, and propionate, serve as a vital energy source for colonocytes and are directly involved in cell differentiation [62]. They can improve epithelial permeability by strengthening the intestinal barrier and can promote ion absorption [63] by decreasing the pH in the intestinal environment [63,64,65,66]. SCFAs have been associated with the alleviation of various diseases and conditions, including obesity, hypertension, autism, colorectal cancer, and cardiovascular disease [67,68]. Our findings indicate that Ramadan fasting leads to a loss of SCFA-producing microbes and a significant reduction in metabolic function, which is crucial for human health. Interestingly, these results are consistent with changes in SCFAs, including acetic acid, butyric acid, and propionic acid. Although the limited sample size for the SCFA assessment did not allow for the calculation of statistical *p*-values among the groups, the apparent decrease in the three SCFAs during the Ramadan fasting period and their return to normal levels after refeeding confirm our findings and also contradict previous studies [11,12,13,14,15,16]. Furthermore, the association between the three assessed SCFAs and microorganisms has helped us understand the health-related roles of microbial communities. For example, the strong positive correlation between *Lactobacillus* and SCFAs not only confirms the previously established role of *Lactobacillus* in improving the intestinal environment but also highlights the importance of the Ramadan dietary strategy for Muslims [69]. The remarkable association between the microorganisms affected by Ramadan fasting and SCFAs also allowed us to gain greater insight into microbial communities. *Bacteroides*, *Fusicatenibacter*, and *Sutterella*, which have a highly positive correlation of over 0.65 with each other, are all strictly or facultatively anaerobic, commensal bacteria in the intestinal environment, and are widely known to produce SCFAs, such as *Lactobacillus* [50,53,54]. Additionally, the increase in fiber intake, which is the primary energy source for SCFA-producing anaerobic microorganisms, can be emphasized even more during Ramadan, when the total calorie intake is restricted.

There are several limitations to the current study due to various external and internal conditions. First, the fecal collection and survey were conducted during the COVID-19 pandemic, and some Islamic participants were unable to complete the questionnaire. As this study serves as a pilot study, with the presence of a baseline group and dietary information in PR2, it would have been possible to explore the relationship more thoroughly between the regimen and the gut microbiome. Additionally, the identification of microorganisms at the genus level using 16S rRNA sequencing may not accurately capture the full range of microorganisms and functional genes correlated with SCFAs, and further studies using shotgun metagenomic sequencing may be necessary. However, despite these limitations, the current study provides unprecedented insights into the effects of Ramadan fasting through a detailed investigation of the changes in metabolites and the gut microbiome. In conclusion, our findings suggest that the health of Ramadan fasting participants may be associated with significant disruptions to the gut microbiota, including the loss of microbial diversity, ASVs, and SCFA-producing bacteria. This novel understanding of microbial perturbation and shifts in metabolites highlights the potential benefits of interventions, such as increasing fiber intake during Ramadan fasting, and emphasizes the importance of further research on the gut microbiome in relation to Muslim health.

## Figures and Tables

**Figure 1 microorganisms-11-02106-f001:**
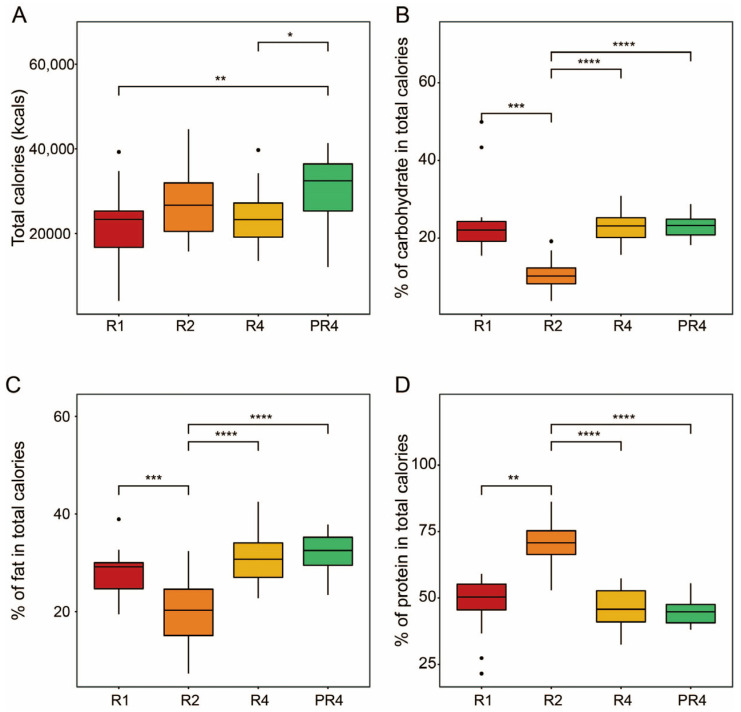
Total calories and macronutrient intake during and post-Ramadan. (**A**) Total calories. Proportions of carbohydrates (**B**), fat (**C**), and protein (**D**). The Friedman test was applied to determine the significance of multiple comparisons, and Dunn’s test was employed for post-hoc analysis; * *p* < 0.05, ** *p* < 0.01, *** *p* < 0.001, **** *p* < 0.0001.

**Figure 2 microorganisms-11-02106-f002:**
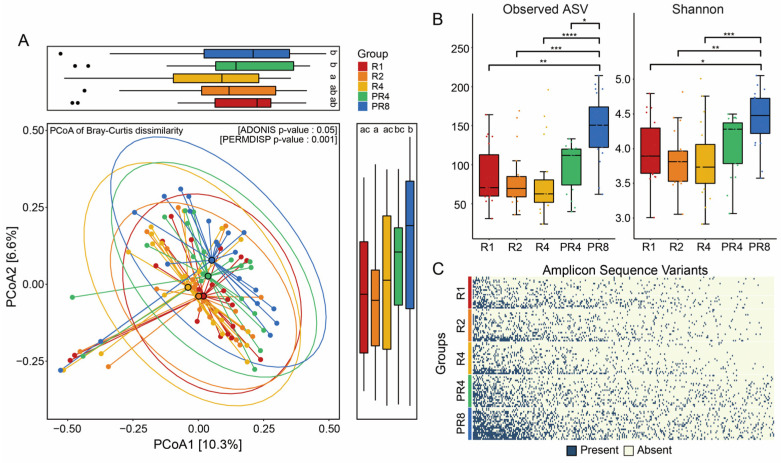
Ramadan fasting induces significant alteration in gut microbial diversity. (**A**) Dissimilarity of gut microbiome through analysis of beta diversity. Principal coordinates analysis of microbial communities based on Bray–Curtis between Ramadan and post-Ramadan (Adonis, *p* = 0.05), including distance of gut microbiome at two axes. The statistical significance among the 5 groups was described using alphabet (Holm correction, *p* < 0.05). (**B**) Dynamic changes in alpha diversities following Ramadan. Friedman’s tests for multiple comparisons and *p*-values were adjusted using the Holm–Bonferroni method for post-hoc test; * *p* < 0.05, ** *p* < 0.01, *** *p* < 0.001, **** *p* < 0.0001. (**C**) Prevalence of ASVs during and post-Ramadan. ASVs presented are sorted by microbial richness within groups and filtered with less than 90% of individual samples.

**Figure 3 microorganisms-11-02106-f003:**
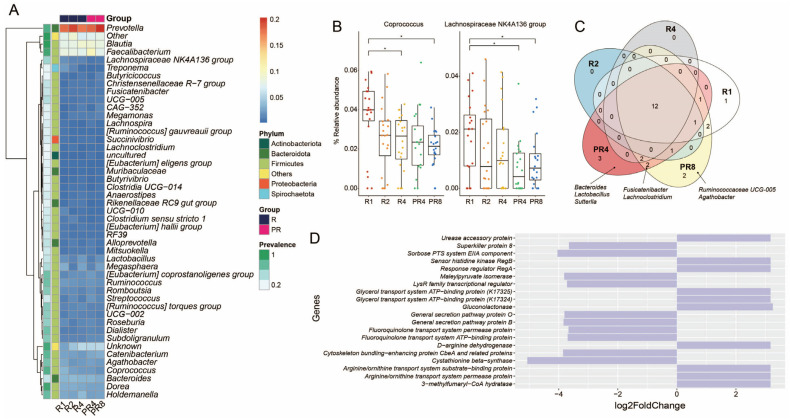
Ramadan model of gut microbiota and metabolic functions. A comparison of the microbial and functional gene profiles was conducted between the groups observing Ramadan (R1, R2, and R4) and those post-Ramadan (PR4 and PR8). (**A**) Heatmap showing the distribution of the genera that were over at least 10% of the relative abundance and 90% of prevalence. (**B**) Tracking the taxa that had statistical significance after Ramadan (Wilcoxon test with Benjamini–Hochberg post-hoc adjustment, * *p* < 0.05). (**C**) Core, accessory, and unique ASVs among 5 groups comprising 3 Ramadan and 2 post-Ramadan groups. (**D**) Bar plot illustrating functional metabolic prediction. Comparison conducted between R4 and PR2 (FDR adjusted *p* < 0.05).

**Figure 4 microorganisms-11-02106-f004:**
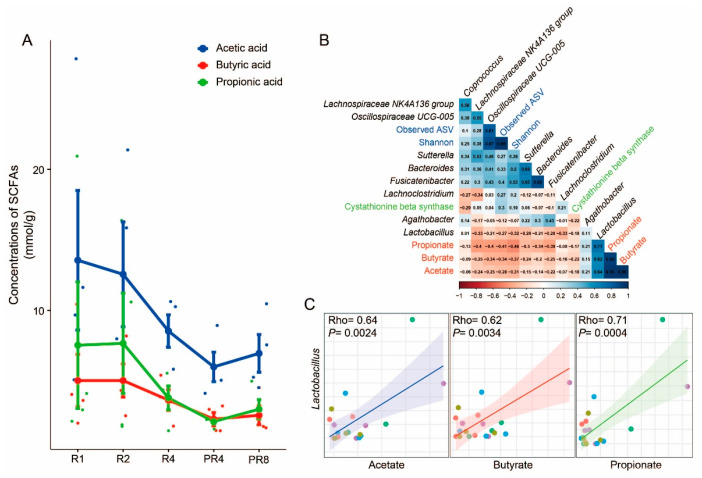
Alterations in SCFAs and correlations with specific taxa. (**A**) The results of SCFA assessment. Acetate, butyrate, and propionate showed a reducing pattern during Ramadan and rebounded at PR8. (**B**) Correlations among assessed SCFAs, microbial diversity, specific taxa, and genes. Each variable is described using a different color. (**C**) Scatter plots with linear correlation between *Lactobacillus* and SCFAs (Rho = 0.64, *p* = 0.0024 for acetate; Rho = 0.62, *p* = 0.0034 for butyrate; Rho = 0.71, *p* = 0.0004 for propionate).

## Data Availability

All of the raw 16S rRNA gene sequence data for this current study were deposited with the National Center for Biotechnology Information’s Bio Project under accession number PRJNA927276.

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
