# Peer review of "The Alteration of the Gut Microbiome during Ramadan Offers a Novel Perspective on Ramadan Fasting: A Pilot Study"

_microorganisms, 2023, doi:10.3390/microorganisms11082106_

Round 1

Reviewer 1 Report

Dear authors,

I revised the manuscript entitled ' The Alteration of the gut Microbiome during Ramadan Offers a Novel Perspective on Ramadan Fasting: A pilot study' which presents a topic of actuality and deserves to be considered for publication in the journal 'microorganisms'. Still, I think the present form of the manuscript must be improved.

General comment: In order to make assumptions about the effects of an intervention, it is important to have a baseline against which all other values can be compared. Therefore, it would have been good to take a stool sample shortly before the start of the Ramadan fast. This must therefore be taken into consideration when interpreting the results of this study. For example the following statement needs to be taken with caution as there is no possibility to compare all the 5 stool samples to a baseline:

·        “In addition, it was confirmed that regardless of fasting, the majority of participants' bacterial composition comprised Prevotella, Blautia, Faecalibacterium, and others (Figure 3A).” (line 236-238)

Please give more information about the sampling of the stool samples:

·        Were the stool samples collected at the test center or did participants collect them at home? If samples were collected at home: How were the samples handled from collection to handing them to the stuff at the study center? What was the min. and max. time between collection of stool samples and storage at -80°C. Were the samples cooled during that time?

·        Was a stabilizer included in the collection tubes? (If so which one?)

·        Since circadian rhythms have a profound impact on the gut microbial composition, it would be of interest to know at which time of day stool samples were collected.

Please add following information about the study participants into the materials and methods part:

·        Sex, age and BMI of the participants

·        Exclusion criteria for the recruitment process

·        Where were the participants recruited?

The introduction covers the main important aspects. Still, there are some points that should be taken into consideration:

Please be more specific on the following phrases and provide the appropriate scientific references supporting these statements:

·        “Dietary changes have effects on gut microbiota which is known to play a key role on host health.” (line 52-53)

·        „One such change that has been extensively studied is intermittent fasting, which has been shown to promote the growth of beneficial bacteria while reducing the abundance of pathogenic bacteria.“ (Line 53-55)

There are studies that indicate that the diversity or overall composition of the gut microbiota does not change significantly after a fasting intervention (or even suggest a decrease of alpha-diversity):

·        DOI: 10.1177/0260106020910907

·        DOI: 10.3390/foods11223673

Please clarify:

·        “Nonetheless, after Ramadan, the population of these taxa has dropped substantially.” (Line 235-236) à Which taxa are meant here?

Please check the correctness of the English language here:

·        “All volunteers were received Written informed consent under the Helsinki Declaration prior to recruitment.” (line 84-85)

Author Response

Dear Reviewers,

We are sincerely thankful to the reviewers for reviewing the manuscript and valuable feedback. We totally agree with all comments provided and have revised the manuscript.

Following this letter are the comments from the reviewers along with our responses, including how and where the text was modified. All the changes made in the revised manuscript are marked using “Track Changes”. The revision was made in consultation with all contributing coauthors, and all of them have approved the revised manuscript. Please find below our point by point responses to the reviewers’ comments.

Thank you for your consideration.

Reviewer 2 Report

Jo and coauthors have studied the gut microbial community structures for the Ramadan fasting participants. Their data showed that Ramadan fasting has significantly disrupted the gut microbiota, including the loss of microbial diversity, ASVs, and SCFA-producing bacteria. The SCFAs increased after Ramadan fasting has been reported before with the shifting of the gut microbiome, thus the novelty of this manuscript is doubted. 

Additionally, there is a big issue in the designation of this study: no fecal specimen before the Ramadan fasting was taken, which makes the conclusion of the manuscript not established. 

In Figure 1, the location of the Y-axis title is wired at the two lower panels, the authors may want to modify it. 

Author Response

(The authors gave the same response as above.)

Round 2

Reviewer 2 Report

Even though there is shortage of experimental design in this study, the results is sufficient and make a conclusion can help audience to understand the Ramadan fasting effects on human gut microbiome and nutrition. 
I think this paper is hitting the publish standard.